# Chained Deep Learning Using Generalized Cross-Entropy for Multiple Annotators Classification

**DOI:** 10.3390/s23073518

**Published:** 2023-03-28

**Authors:** Jenniffer Carolina Triana-Martinez, Julian Gil-González, Jose A. Fernandez-Gallego, Andrés Marino Álvarez-Meza, Cesar German Castellanos-Dominguez

**Affiliations:** 1Signal Processing and Recognition Group, Universidad Nacional de Colombia, Manizales 170003, Colombia; jectrianama@unal.edu.co (J.C.T.-M.);; 2Department of Electronics and Computer Science, Pontificia Universidad Javeriana Cali, Cali 760031, Colombia; 3Programa de Ingeniería Electrónica, Facultad de Ingeniería, Universidad de Ibagué, Ibagué 730001, Colombia

**Keywords:** deep learning, multiple annotators, chained approach, generalized cross-entropy, classification

## Abstract

Supervised learning requires the accurate labeling of instances, usually provided by an expert. Crowdsourcing platforms offer a practical and cost-effective alternative for large datasets when individual annotation is impractical. In addition, these platforms gather labels from multiple labelers. Still, traditional multiple-annotator methods must account for the varying levels of expertise and the noise introduced by unreliable outputs, resulting in decreased performance. In addition, they assume a homogeneous behavior of the labelers across the input feature space, and independence constraints are imposed on outputs. We propose a Generalized Cross-Entropy-based framework using Chained Deep Learning (GCECDL) to code each annotator’s non-stationary patterns regarding the input space while preserving the inter-dependencies among experts through a chained deep learning approach. Experimental results devoted to multiple-annotator classification tasks on several well-known datasets demonstrate that our GCECDL can achieve robust predictive properties, outperforming state-of-the-art algorithms by combining the power of deep learning with a noise-robust loss function to deal with noisy labels. Moreover, network self-regularization is achieved by estimating each labeler’s reliability within the chained approach. Lastly, visual inspection and relevance analysis experiments are conducted to reveal the non-stationary coding of our method. In a nutshell, GCEDL weights reliable labelers as a function of each input sample and achieves suitable discrimination performance with preserved interpretability regarding each annotator’s trustworthiness estimation.

## 1. Introduction

Conventional Machine Learning (ML) and Deep Learning (DL) techniques utilize a prediction function that maps input data to output targets. In supervised tasks, output values (or “ground truth”) are available for training, but in many real-world scenarios, these values may be unknown or too costly to obtain [1]. With the rise of DL-based approaches, there has been an increasing interest in their use as the primary tool in various classification and regression tasks [2]. However, a crucial factor that dramatically impacts the performance of DL models is the quantity and quality of labeled data used during training [3]. Concerning this, crowdsourcing is a widely recognized approach for obtaining labeled data cost-effectively and efficiently from multiple annotators. It involves the use of online platforms, such as Amazon Mechanical Turk (AMT), to recruit a large number of individuals (annotators) to label the data and provide their subjective interpretation of the unknown ground truth [4].

In the ML field, assigning labels to instances with the help of multiple annotators is a common practice. However, it presents a significant challenge when traditional supervised algorithms are applied because they rely on the assumption that the training labels provided by a single expert are reliable [5]. When multiple annotators with varying levels of expertise are employed, the reliability of the labels becomes uncertain, leading to decreased performance and inaccurate model predictions. Addressing this issue is crucial for developing practical ML and DL models that perform well on real-world datasets.

Traditional supervised learning algorithms cannot account for the varying levels of expertise and the noise introduced by unreliable labels, resulting in decreased performance. Further research is needed to develop methods that can effectively handle the problem of multi-annotator label aggregation and overcome this challenge. Moreover, several issues arise when using DL in a multiple annotator scenario. One of the main challenges is the variability of annotator effectiveness, which may depend on the sample instance presented. Even for a single task, annotators can provide inconsistent or incorrect outputs, leading to noisy labels [6]. These samples can negatively impact the model’s performance by affecting the gradients and making it difficult to converge on a suitable solution [7,8].

Learning from Crowds (LFC) scenarios pose a significant challenge for ML models, and multiple approaches have been developed to tackle this issue. The most commonly used strategy is to adapt supervised learning algorithms and use majority voting for label aggregation. Yet, this approach has limitations since it assumes that annotators have the same level of reliability [9,10]. In addition, incorrect labels and outliers can influence the consensus, decreasing performance. Therefore, more advanced models, such as the Expectation-Maximization (EM) framework, have been considered to address these issues [11,12]. This approach simultaneously estimates true labels and annotator reliability, making more accurate predictions. Another strategy is to train a supervised learning algorithm while also modeling annotator behavior [13]. This approach yields better results than label aggregation and can be used to identify unreliable labelers and remove their outputs from the training set. On the other hand, recent work has shown that relaxing the independence assumption among annotators can lead to more accurate ground truth estimation [14,15]. Then, sophisticated models have been proposed, such as those that use regression tasks to model annotator behavior employing a multivariate Gaussian distribution [16,17]. Moreover, such techniques can help identify the relationships among experts and improve the overall accuracy of the predictions.

Furthermore, learning with noisy labels is a challenging problem in ML and DL. Recently, numerous methods have been proposed for learning noisy labels with DL-based approaches [18]. These methods can involve developing a robust architecture [19,20], enforcing a DL-model with robust regularization techniques [8] and identifying true labeled examples from noisy training data via multi-network [21,22] or multi-round learning [23,24]. In addition, sparse loss functions such as the Mean Absolute Error (MAE) are employed [25,26]. The MAE can help the model focus on correctly labeled examples; however, performance decreases when it is used on large and complex databases [26].

Here, we introduce a Chained Deep Learning (CDL) strategy to learn from multiple noisy annotators in classification tasks. Such an approach allows coding the non-stationary patterns of each annotator regarding the input space while preserving the inter-dependencies among experts. In addition, we combine the capabilities of CDL with a Generalized Cross-Entropy-based loss function aiming to build a model, termed the Generalized Cross-Entropy-based Framework using CDL (GCECDL), that is less prone to outlier annotations. Our proposal is similar to the works in [27,28] in that we use a deep-learning-based strategy to build a supervised learning model in the context of multiple annotators. Moreover, network self-regularization is accomplished by predicting each labeler’s reliability within our chained scheme. On the other hand, the proposed research uses t-distributed Stochastic Neighbor Embedding- (t-SNE) [29] and Gradient-based Class Activation Maps [30] to interpret and validate the obtained results visually. Finally, experimental results related to multiple-annotator classification on several well-known datasets (synthetic and real-world scenarios) demonstrate that GCECDL outperforms state-of-the-art techniques.

The agenda for this paper is as follows: Section 2 summarizes the related work. Section 3 describes the methods. Section 4 and Section 5 present the experiments and discuss the results. Finally, Section 6 outlines the conclusions and future work.

## 2. Literature Review

Several approaches have been developed to address LFC scenarios. In this light, we recognize two main groups: combining the annotations to estimate the gold standard or adapting supervised learning algorithms to that type of labels [31]. The primary method is called label aggregation. One of the most used techniques is known as Majority Voting (MV), which has been applied to different multi-labeler problems due to its simplicity [32]. In MV, the most frequent output among the experts is chosen as the final prediction. The latter is simple to implement and effective in some cases, but it also has limitations because MV relies on the assumption that annotators have the same level of reliability, which is challenging to fulfill in real-world scenarios. Additionally, the consensus can be heavily influenced by incorrect labels and outliers [10]. In this sense, EM methods have been proposed to handle imbalanced labeling to handle these issues [32,33]. The EM framework aims to estimate the true label and the annotator’s reliability simultaneously, while methods to handle imbalanced labeling try to adjust for the differences in the annotator’s expertise level. These more advanced models provide more robust solutions to the problem of multi-annotator label aggregation and can lead to better performance than MV.

An alternate approach to addressing multi-labeler tasks is to simultaneously train a supervised learning algorithm while also modeling the behavior of the annotators. It has been empirically demonstrated that this approach yields better results than label aggregation. Furthermore, features that train the learning algorithm can also be exploited to infer the ground truth [13]. One introductory study in this field is the EM-based framework presented by authors in [34], which estimates the sensitivity and specificity of annotators while also training a logistic regression classifier. This approach has served as a foundation for various algorithms that address multi-labeler tasks, including regression [35,36], binary classification [37,38], multi-class classification [4,39], and sequence labeling [40]. Likewise, some studies have adapted these ideas for DL techniques by incorporating an additional layer that contemplates multiple labelers [27,28].

In turn, the study in [15] represents an early exploration of the relationship between annotators’ parameters and input features. The authors propose a method for binary classification with multiple labelers, where input data are grouped using a Gaussian Mixture Model (GMM). The algorithm posits that annotators have specific performance levels in terms of sensitivity and specificity. However, it does not incorporate information from multiple experts as input for GMM, which may result in labelers’ parameter deviation. Likewise, the authors in [6] presented a binary classification algorithm that employs Bernoulli and Gaussian distributions to code the annotators’ performance as a function of the input space. In addition, a linear relationship between the annotator’s expertise and the input space is assumed, which can be problematic. For example, when assessing documents online, annotators may have varying levels of labeling accuracy. These differences could be due to their familiarity with specific topics related to the studied documents [41]. Moreover, in [36], a Gaussian Process (GP)-based regression procedure is proposed to incorporate multiple annotators. The annotators’ parameters are estimated as a nonlinear function of the input space by using an additional GP. Nevertheless, since this approach is based on a classical formulation of GPs, its computational complexity is prohibited for large datasets [39]. Furthermore, relaxing the independence assumption among annotators has led to a more accurate estimation of the ground truth, as demonstrated in [14,15]. In [17], an unsupervised regression task is described where the labelers’ behavior is modeled using a multivariate Gaussian distribution, with the covariance matrix encoding the annotators’ interdependencies. Again, the authors in [38] proposed a binary classification method that utilizes a weighted combination of classifiers. The weights are estimated using a kernel alignment-based algorithm.

Of note, when multiple annotators are present, the training set may contain noisy labels, negatively impacting the model’s generalization ability. Typical regularization approaches, such as dropout and batch normalization, only partially outweigh the over-fitting drawback for DL [18]. An alternative is to use more robust loss functions to train DL models. Because of its fast convergence and generalization capability, most deep learning-based classifiers use Categorical Cross-Entropy (CE) as cost function. Nevertheless, MAE has been found to perform better when dealing with noisy labels [26]. However, the robustness of MAE can concurrently cause increased difficulty in training and lead to a performance drop. This limitation is particularly evident when using neural networks on complicated datasets. To combat this drawback, Zhang et al. [42] proposed the GCE, establishing a more general type of noise-robust loss function taking advantage of both MAE and CE, yielding good performance in the presence of noisy labels. Moreover, it can be readily applied to any existing neural network architecture.

Our proposal follows the lines of the work in [27,28] in that GCECDL uses a deep-based approach to build a supervised learning model in the context of multiple-annotator classification. Yet, while such approaches code the annotators’ parameters as fixed points, we model them as functions to consider dependencies between the input features and the labelers’ behavior. GCECDL is also similar to the works in [14,43]. Both approaches model the annotators’ performance as a function of the input instances and consider the interdependencies among the labelers. Even so, unlike [14], where it is necessary to use as many classifiers as annotators, our approach only needs to train a single classifier from a DL representation, which is advantageous for a large number of labelers. Moreover, unlike [43], our loss function can deal with noisy labels and more difficult training scenarios by using a generalization between MAE and CE. Indeed, network self-regularization is accomplished by predicting each labeler’s reliability.

## 3. Methods

### 3.1. Generalized Cross-Entropy

Let us consider a *K*-class classification problem from a given prediction function $f:R^{P}\to {\left[0,1\right]}^{K}$, trained on the input–output set $\left\{X\in R^{N\times P},Y\in {\left\{0,1\right\}}^{N\times K}\right\}$ and holding P—dimensional input features in *N* row vectors $x_{n}\in R^{P}$ corresponding to each ground truth label $y_{n}\in {\left\{0,1\right\}}^{K}$, n∈{1,2,…,N}. The prediction function f(·) is commonly coupled by a softmax output to fulfill 1−K one-hot labels. In turn, the well-known Mean Absolute Error (MAE) and Categorical Cross-Entropy (CE) losses, used typically for optimizing *f*, are defined as follows: (1)$$\begin{array}{cc} MAE(y,f(x))= & {∥y-f\left(x\right)∥}_{1}, \end{array}$$(2)$$\begin{array}{cc} CE(y,f(x))= & \sum_{k=1}^{K}y_{k}\log\left(f_{k}\left(x\right)\right); \end{array}$$
where $y_{k}\in y,$$f_{k}\left(x\right)\in f\left(x\right)$, and ${∥\cdot ∥}_{1}$ stands for the $l_{1}$-norm. Of note, $1^{⊤}y=1^{⊤}f\left(x\right)=1,$
1∈${\left\{1\right\}}^{K}$ being an all-ones vector. In addition, the MAE loss can be rewritten for softmax outputs, yielding:(3)$$MAE\left(y,f\left(x\right)\right)=2(1-1^{⊤}\left(y⊙f\left(x\right)\right))$$
where ⊙ stands for the Hadamard product.

On the one hand, CE is sensitive to label noise, being a nonsymmetric and unbounded loss function. On the other hand, MAE is noise-robust because of its symmetric property, that is [26]:(4)$$\sum_{k=1}^{K}MAE(y,f(x)|k)=\sum_{k=1}^{K}2(1-f_{k}\left(x\right))=C,\;\forall x\in R^{P},\forall f;$$
where $C\in R^{+}.$ The symmetric property of MAE for softmax-based outputs allows extending the L1-norm expression in Equation Equation 1 to a vectorized form in Equation Equation 3. Note that the symmetric property is only fulfilled for softmax-based representations. Therefore, the L1-norm favors sparse coding when computing the mismatch between target and prediction, favoring the filtering of noisy outputs, as commonly studied for L1-based filtering approaches [44].

Though MAE is symmetric and bounded, it also has some drawbacks when used as classification loss for deep learning networks trained on large datasets employing stochastic gradient-based techniques. Specifically, for a given network with parameter set θ, the MAE and CE gradients can be computed as: (5)$$\begin{array}{cc} \frac{\partial MAE(y,f(x;\theta )|k)}{\partial \theta }= & -\nabla_{\theta }f_{k}\left(x;\theta \right), \end{array}$$(6)$$\begin{array}{cc} \frac{\partial CE(y,f(x;\theta )|k)}{\partial \theta }= & -\frac{1}{f_{k}\left(x;\theta \right)}\nabla_{\theta }f_{k}\left(x;\theta \right). \end{array}$$

As seen in Equations Equation 5 and Equation 6, less congruent samples have greater weights in CE than predictions that agree more with ground truth labels; meanwhile, the MAE penalizes equally during gradient descent optimization. At first glance, MAE can deal with noisy labels; still, this can lead to longer and more difficult training scenarios, particularly for large databases.

Therefore, authors in [42] proposed a trade-off between MAE and CE using a Box–Cox transformation [45], yielding to the following Generalized Cross-Entropy (GCE) loss for training deep learning models:(7)$$GCE\left(y,f\left(x\right)\right)=2\frac{1-{\left(1^{⊤}\left(y⊙f\left(x\right)\right)\right)}^{q}}{q},$$
with q∈(0,1]. Remarkably, the limiting case for q→0 in GCE is equivalent to the CE expression, and when q=1, it equals the MAE loss. In addition, the GCE in Equation Equation 7 holds the following gradient with regard to θ:(8)$$\frac{\partial GCE(y,f(x;\theta )|k)}{\partial \theta }=-f_{k}{\left(x;\theta \right)}^{q-1}\nabla_{\theta }f_{k}\left(x;\theta \right).$$

As depicted in Equation Equation 8, the GCE’s gradient weighs samples using the $f_{k}{\left(x;\theta \right)}^{q-1}$ factor, which could affect robustness against noisy labels depending on the hyperparameter value *q*. In summary, the larger the *q* value, the more noise robustness is attained. Therefore, a suitable *q* is required to find a trade-off between noisy robustness and better learning dynamics during network training.

Lastly, since tighter loss bounding would imply more robust noise tolerance, GCE can be extended to its truncated version as follows [42]:(9)$$TGCE\left(y,f\left(x\right);{\tilde{\lambda }}_{x},\tilde{C}\right)={\tilde{\lambda }}_{x}\frac{1-{\left(1^{⊤}\left(y⊙f\left(x\right)\right)\right)}^{q}}{q}+\left(1-{\tilde{\lambda }}_{x}\right)\frac{1-{\left(\tilde{C}\right)}^{q}}{q},$$
where $\lambda_{x}\in$ [0,1] and $\tilde{C}\in$(0, 1). Note that $\lambda_{x}$ prunes samples regarding a noise tolerance ruled by $\tilde{C}$.

### 3.2. Chained Deep Learning Fundamentals

The seminal Chained Gaussian Processes (CGP) in [46] fixes a likelihood function with *J* parameters depending on the input–output set {X,Y}, as follows:(10)$$\begin{array}{c} p\left(Y|X,\xi \right)=\prod_{n=1}^{N}p\left(y_{n}|\xi_{1}\left(x_{n}\right),\ldots ,\xi_{J}\left(x_{n}\right)\right), \end{array}$$
$\xi \;=\;{\left[\xi_{1}\ldots ,\xi_{J}\right]}^{⊤}\in R^{NJ}$ being a parameter vector and $\xi_{j}\;=\;{\left[\xi_{j}\left(x_{1}\right)\ldots \xi_{j}\left(x_{N}\right)\right]}^{⊤}\in R^{N}$. In addition, each $\xi_{j}\left(x\right)\in M_{j}$ maps an input instance to the parameter space ($j\in \left\{1,2,\ldots ,J\right\}$). The likelihood in Equation Equation 10 allows modeling the function parameters with *J* independent GPs (one GP prior per parameter).

Likewise, we can extend the concept of CGP to the field of DL. Hence, suppose a DNN with *L* layers, where the output layer contains *J* outputs (neurons). The DNN model can be represented by the following composite function $f\left(x\right)={\left[f_{1}\left(x\right),\ldots ,f_{J}\left(x\right)\right]}^{⊤}\in R^{J},$
(11)$$\begin{array}{c} f\left(x\right)=\left(\varphi_{L}\circ \varphi_{L-1}\circ \cdots \circ \varphi_{1}\right)\left(x\right), \end{array}$$
where ∘ stands for function composition. Accordingly, Chained Deep Learning (CDL) links each likelihood parameter $\xi_{j}\left(x\right)$ to one of the *J* outputs. Each CDL parameter can be estimated as: $\xi_{j}\left(x\right)=h_{j}\left(f_{j}\left(x\right)\right)$, where $h_{j}:R\to M_{j}$ maps each $f_{j}\left(x\right)$ prediction to $M_{j}$. Furthermore, each function $\varphi_{l}\left(\cdot \right)$, with $l\in \left\{1,2,\ldots ,L\right\}$, depends on a set of parameters $\phi ={\left[\phi_{1},\ldots ,\phi_{L}\right]}^{⊤}$, e.g., weights and biases, that can be optimized via gradient descent and back-propagation [47].

### 3.3. Generalized Cross-Entropy-Based Chained Deep Learning for Multiple Annotators

Nowadays, in several real-world classification problems, instead of the ground truth Y, multiple labels are provided by *R* experts with different levels of ability, e.g., Multiple Annotators (MA) [28]. For the sake of clarity, we assume that the *r*-th expert annotates $|\Omega_{r}|\leq N$ instances, $|\Omega_{r}|$ being the cardinality of the set $\Omega_{r}$ containing the indices of samples labeled by annotator *r*. Moreover, let $\Psi_{n}$ be the index set gathering the annotators who labeled the *n*-th instance. Then, a multiple annotators dataset $\{X\in R^{N\times P}$, $\tilde{Y}\in {\left\{0,1,⌀\right\}}^{N\times K\times R}\}$, where ${\tilde{y}}_{n}^{r}\in {\left\{0,1,⌀\right\}}^{K}$ is the 1−K one-hot label of expert *r* for instance *n*, can be built to feed a CDL approach holding J=R+K outputs. The former *R* outputs code each expert reliability $\lambda_{n}^{r}\in \left\{0,1\right\}$, and the remaining *K* predictions approximate the ground truth $y_{n}$.

In this sense, given an input sample x, each annotator’s reliability can be predicted by fixing a sigmoid activation to the first *R* neurons within layer *L* in Equation Equation 11, as:(12)$${\hat{\lambda }}_{n}^{r}=\frac{1}{1+\exp(-f_{r}\left(x_{n}\right))},$$
where ${\hat{\lambda }}_{n}^{r}\in \left[0,1\right].$

Moreover, a softmax function is set to the last *K* outputs in $\phi_{L}\left(\cdot \right)$ to predict the ground truth label, as follows:(13)$${\hat{y}}_{k}=\frac{\exp(f_{R+k}\left(x\right))}{\sum_{i=R+1}^{J}\exp\left(f_{i}\left(x\right)\right)}.$$
where k={1,2,…,K},
${\hat{y}}_{k}\in \left[0,1\right]$, and $\sum_{k}{\hat{y}}_{k}=1.$

Here, to circumvent noisy annotators while coding their non-stationary behavior along the input space, and to favor the CDL training ruled by the optimization of the parameter set ϕ, a TGCE-based loss as in Equation Equation 9 is proposed for multiple-annotator classification, yielding:(14)$$\phi^{*}=\arg\min_{\phi }\sum_{n=1}^{N}\sum_{r\in \Psi_{n}}\left[{\hat{\lambda }}_{n}^{r}\left(\phi \right)\sum_{k=1}^{K}{\tilde{y}}_{n,k}^{r}\left(\frac{1-{\hat{y}}_{n,k}^{q}\left(\phi \right)}{q}\right)+\left(1-{\hat{\lambda }}_{n}^{r}\left(\phi \right)\right)\left(\frac{1-{\left(\frac{1}{K}\right)}^{q}}{q}\right)\right]$$
where ${\tilde{y}}_{n,k}^{r}\in {\tilde{y}}_{n}^{r}$ and q∈(0,1].

As seen above, self-regularization is achieved through each expert’s reliability estimation ${\hat{\lambda }}_{n}^{r}\left(\phi \right)$ in Equation Equation 14, which prunes the TGCE loss ruled by *q*. Of note, when ${\hat{\lambda }}_{n}^{r}\left(\phi \right)\to 1,$
${\hat{y}}_{n}\left(\phi \right)\in {\left[0,1\right]}^{K}$ holds the 1-K ground truth predictions as in Equation Equation 13. As a consequence, only samples with ${\hat{\lambda }}_{n}^{r}\left(\phi \right)\to 1$ are kept for updating the CDL parameters. In contrast, noisy or unreliable annotations (${\hat{\lambda }}_{n}^{r}\left(\phi \right)\to 0$) are avoided to update the network parameters. Therefore, our GCE-based CDL, termed GCECDL, allows coding the non-stationary patterns of each annotator regarding each input instance space while preserving the interdependencies among experts through a CDL approach. In summary, our approach benefits CDL and GCE by circumventing noisy experts with non-stationary patterns. Figure 1 summarizes the GCECDL sketch.

## 4. Experimental Set-Up

The following section comprehensively describes the tested datasets and the key experimental conditions utilized.

### 4.1. Tested Datasets

Our GCECDL approach, designed for multiple-annotator classification, is evaluated using synthetic and real-world datasets. The experiments aim to uncover the key insights and advantages of GCECDL for coding non-stationary and unreliable expert labels on complex datasets.

We generate synthetic data for a 1-dimensional, 3-class classification problem by randomly sampling 5000 points from a uniform distribution within the interval [0,1] and using these points to construct the input feature matrix X. The true label $y_{n,k}$ for each sample is determined by taking the maximum value of $t_{n,k}$ for *k* in the set 1, 2, 3, where $t_{n,1}=\sin\left(2\pi x_{n}\right)$, $t_{n,2}=-\sin\left(2\pi x_{n}\right)$, and $t_{n,3}=-\sin\left(2\pi \left(x_{n}+0.25\right)\right)+0.5$. We also create a test set by extracting 2000 equally spaced samples from the same interval.

We then look for datasets where the input data come from real-world applications. Still, the labels from multiple annotators are obtained synthetically. The synthetic labeling is carried out to control the labeling process. In particular, six binary and multi-class classification task datasets are studied from the famous UCI repository (http://archive.ics.uci.edu/ml, accessed on 19 August 2022). The chosen datasets include: Occupancy Detection (Occupancy), Skin Segmentation (Skin), Tic-Tac-Toe Endgame (tic-tac-toe), Iris Plants (Iris), Wine (Wine), and Image Segmentation (Segmentation). Moreover, the Fashion-MNIST dataset [48], as well as the Balance and New Thyroid datasets, are also selected from the Keel-dataset Repository (https://sci2s.ugr.es/keel/category.php?cat=clas, accessed on 3 October 2022). In addition, the publicly available bearing data collected by the Case Western Reserve University (Western) are used. The aim is to build a system to diagnose an electric motor’s status based on two accelerometers. The feature extraction is performed as in [49]. We also evaluate our proposed GCECDL classifier on two large image classification sets: MNIST of Handwritten Digits (MNIST) [50], an easily interpretable image database of labeled handwritten digits with 60,000 images for training and 100,000 for test sets, and the Cats vs. Dogs database, consisting of 25,000 images of dogs and cats [51], each class being represented by 1 and 0, respectively.

Finally, we include three real-world datasets provided with human annotations. First, the CIFAR-10H comprises over 500 k crowdsourced human categorization judgments obtained through AMT and includes ten categories: airplane, automobile, bird, cat, deer, dog, frog, horse, ship, and truck [52]. In our study, we applied a rigorous data-filtering process and discarded any samples that at least one annotator did not label. This filtering step resulted in 19.233 labeled samples, out of which 10.000 were used for testing. The second dataset is LabelMe, which aims to classify images into eight different classes: highway, inside city, tall building, street, forest, coast, mountain, and open country. It consists of 2688 images; each image was labeled by an average of 2547 workers, with a mean accuracy of 69.2%. We used the prepared dataset from [28], which performs a feature extraction stage based on a pre-trained VGG-16 deep neural network [53]. The third one is the Music genre database [54], comprising one thousand 30-second samples of songs categorized into classical, country, disco, hip-hop, jazz, rock, blues, reggae, pop, and metal. Each class contains 100 representative samples. A random selection of 700 samples was published on the AMT platform for workers to classify them from one to ten based on their genre. Feature extraction was performed following the method outlined in [55], resulting in an input space of 124 features. Table 1 summarizes the tested synthetic, semi-synthetic, and real-world datasets.

### 4.2. Provided and Simulated Annotations

To test our GCECDL classifier, we simulate annotator labels as corrupted versions of the hidden ground truth. Here, the simulations are performed by assuming: (i) dependencies among annotators and (ii) the labelers’ performances are modeled as a function of the input features. In turn, the semiparametric latent factor model is used to build the labels, as follows [56]:Define *Q* deterministic functions ${\hat{\mu }}_{q}:\chi \to R\;$ and their combination parameters ${\hat{\omega }}_{l_{r},q}\in R,\forall r\in R,n\in N$. Compute ${\hat{f}}_{l_{r},n}=\sum_{q=1}^{Q}{\hat{\omega }}_{l_{r},q}{\hat{\mu }}_{q}\left({\hat{x}}_{n}\right)$, where ${\hat{x}}_{n}\in R$ is the *n*-th component of $\hat{x}\in R^{N}$, $\hat{x}$ being the 1-D representation of the input features in X by using t-SNE approach [29].Calculate $\lambda_{n}^{r}=\mathrm{sigmoid}\left({\hat{f}}_{l_{r},n}\right)$, where sigmoid(·)∈[0,1] is the sigmoid function.Finally, find the *r*-th label as $y_{r}^{n}=\left\{\begin{array}{cc} y_{n} & \;\mathrm{if}\lambda_{r}^{n}\geq 0.5\\ {\overset{˘}{y}}_{n} & \;\mathrm{if}\lambda_{r}^{n}<0.5 \end{array}\right.$, where ${\overset{˘}{y}}_{n}$ is a flipped version of the actual label $y_{n}$.


### 4.3. Performance Measures, Method Comparison, and Training

As quantitative assessment concerning the classification performance, the overall Accuracy (ACC) and the Balanced Accuracy (BACC) are reported on the testing set, which can be written as follows:(15)$$\begin{array}{c} ACC\left[\%\right]=100\frac{TP+TN}{TP+TN+FP+FN} \end{array}$$
(16)$$\begin{array}{c} BACC\left[\%\right]=\frac{100}{2}\left(\frac{TP}{TP+FN}+\frac{TN}{TN+FP}\right) \end{array}$$
where TP, FN, and FP represent the true positive, false negative, and false positive predictions, respectively, after comparing the actual and estimated labels $y_{n}$ and ${\hat{y}}_{n}$ for a given input sample $x_{n}$.

In addition, we consider the Normalized Mutual Information (NMI) between the output and the target [57]. The NMI measures the amount of shared information between two variables and quantifies the strength of their relationship, yielding:(17)$$\begin{array}{c} \mathrm{NMI}\left[\%\right]=100\frac{1}{N}\sum_{n=1}^{N}\frac{2I(y_{n},{\hat{y}}_{n})}{H\left(y_{n}\right)+H\left({\hat{y}}_{n}\right)}, \end{array}$$
where I(·,·) stands for mutual information and H(·) for marginal entropy. Furthermore, we estimate the Area Under the ROC Curve (AUC) that can be computed by varying the decision boundary concerning the sensitivity (Sen) and specificity (Spe) measures, as follows [47]:(18)$$\begin{array}{c} AUC\left[\%\right]=100\frac{Sen+Spe}{2} \end{array}$$

For concrete testing, we use a cross-validation scheme with 10 repetitions holding 70% of the samples for training and the remaining 30% for testing (except for the Mnist, F-Mnist, CIFAR-10H, and music dataset, where training and testing sets are clearly defined).

Moreover, Table 2 displays the tested state-of-the-art algorithms for comparison purposes. The abbreviations are fixed as follows: Regularized Chained Deep Neural Network Classifier for Multiple Annotators (RCDNN) [43], Deep Learning Majority Voting (DL-MV), and Deep Learning from Crowds (DL-CW(MW)) [28]. Our Python codes are publicly available for DL-CL(MW), DL-MV, RCDNN, and GCECDL at https://github.com/Jectrianama/GCCE_TEST (accessed on 19 December 2022). Regarding DL-CL(MW), we use the codes at http://www.fprodrigues.com/ (accessed on 19 December 2022). Of note, DL-GOLD is a deep learning model trained with the true label, which is used only to provide an upper bound.

In turn, to better grasp the behavior of our GCECDL classifier over every dataset, we implemented a grid-search scheme to fix the *q* value within the grid [0.001,0.1,0.2,0.3,0.4,0.5,0.75].

The proposed GCECDL architecture for multiple annotators comprises (i) a fully connected network for tabular data (see Figure 2) and (ii) a convolutional network for image data (see Figure 3). For all provided layers, elastic-net-based weight regularizers are used. As usual, the optimization problem is solved using a back-propagation algorithm. Moreover, to favor scalability, we utilize a mini-batch-based gradient descent approach with automatic differentiation (the Adam-based optimizer is fixed). In addition, we employed callbacks during the training process to monitor the model’s performance. Specifically, we used an EarlyStopping callback to stop the training process if the validation performance did not improve for a specified number of epochs and a LearningRateScheduler callback, allowing the model to converge more quickly by avoiding becoming stuck in a suboptimal solution. These callbacks allowed us to optimize the performance of our neural network and sidestep overfitting. Finally, we selected the best performance between models with or without callbacks for each database. All experiments were conducted in Python 3.8, with the Tensorflow 2.4.1 API, on a Google Colaboratory environment.

## 5. Results and Discussion

### 5.1. Reliability Estimation and Visual Inspection Results

We first perform a controlled experiment to test the GCECDL’s capability when dealing with binary and multiclass classification. We use the one-dimensional synthetic dataset described in Section 4. In addition, five labelers (R = 5) are simulated with different levels of expertise. To simulate the error variances, we define Q=3${\hat{\mu }}_{q}\left(\cdot \right)$ functions (see Section 4.2), yielding: (19)$$\begin{array}{cc} {\hat{\mu }}_{1}\left(x\right)= & 4.5\cos(2\pi x+1.5\pi )-3\sin(4.3\pi x+0.3\pi ) \end{array}$$(20)$$\begin{array}{cc} {\hat{\mu }}_{2}\left(x\right)= & 4.5\cos(1.5\pi x+0.5\pi )+5\sin(3\pi x+1.5\pi ) \end{array}$$(21)$$\begin{array}{cc} {\hat{\mu }}_{3}\left(x\right)= & 1, \end{array}$$
where x∈[0,1]. In addition, the combination weights are gathered within the following combination matrix $\hat{W}\in R^{Q\times R}$:(22)$$\hat{W}=\left[\begin{array}{ccccc} 0.4 & 0.7 & -0.5 & 0.0 & -0.7\\ 0.4 & -1.0 & -0.1 & -0.8 & 1.0\\ 3.1 & -1.8 & -0.6 & -1.2 & 1.0 \end{array}\right]$$
holding elements ${\hat{\omega }}_{l_{r},q}$, which are used to combine functions ${\hat{\mu }}_{1}$, ${\hat{\mu }}_{2}$, and ${\hat{\mu }}_{3}$.

For visual inspection purposes, Figure 4 shows the predictive label’s probability (PLP) and the AUC, for all studied approaches regarding the one-dimensional synthetic database.

As seen in Figure 4, DL-CL(MW) and RCDNN have a different shape than the ground truth. Additionally, DL-MV has the worst accuracy for two out of the three classes. Upon further analysis of the results of our GCECDL method, we note that its predictive accuracy is quite close to the absolute ground truth, which is the theoretical upper limit. Thus, GCECDL offers a more suitable representation of the labelers’ behavior compared to its competitors. This is because GCECDL takes into account both the annotators’ dependencies and the relationship between the input features and the annotators’ performance.

To support the previous statement, Figure 5 illustrates the per-annotator reliability estimated by our model and the simulated accuracy of each annotator. As can be seen, our model provides an excellent representation for annotators one and five and an acceptable representation for annotators two, three, and four. This is a direct result of modeling the labelers’ parameters as functions of the input features. This outcome demonstrates how our approach effectively identifies the areas where a specific labeler aligns with the regions of higher accuracy of the simulated annotators.

In this next step, we conduct two crucial experiments utilizing the MNIST dataset, as outlined in Section 4. These experiments include an examination of explainable multiple-annotator classification and a t-SNE-based 2D visualization of the data. To achieve this, we employed the Gradient-weighted Class Activation Mapping (Grad-CAM++) approach to extract normalized class activation mapping from the image data, as described in [30]. We then plotted heatmap images related to the FC 2 layer, which represents the high-level visual features that are extracted from the characteristics. These feature maps are then projected onto a two-dimensional space using the t-SNE algorithm. The visual analysis of these results shows that the same color image samples cluster together in the 2D, low-dimensional space according to their class, while preserving the spatial relationships from the input space.

Figure 6 presents a visualization of the gold standard and simulated annotators (R=3 for illustrative purposes) plotted over the resulting 2D features projection for the training set. It can be observed that the features extracted possess a high degree of separability and discriminative ability, as every class (0–9) is represented by a distinct cluster. For illustration purposes, a few images are depicted over each corresponding projection. The last two rows show a selection of simulated labels and their different scores (annotator reliability). We can see the different levels of expertise obtained from the confusion matrix. The first annotator, whose accuracy score with respect to the ground truth labels is 97%, is depicted over the projection. We can observe how most samples have a correct version of the ground truth. However, it tends to fail more for classes 0, 2, 3, and 8. This behavior becomes more pronounced for the last two annotators, whose accuracy drops to 41% and 11%, respectively. The mismatch between the labels and the ground truth is more evident in the top figure. We expected to compare this with the estimated reliability obtained by our model.

Then, Figure 7 illustrates the hidden ground truth prediction and reliability estimation generated by our GCECDL approach. As shown, GCECDL demonstrates a high level of suitability for the MNIST digit classification problem by achieving an ACC score of 0.99, which highlights its generalization capability, even in cases where the ground truth is unknown. This is because the proposed model takes into account both the relationship between the input space and the annotator’s behavior, as well as the dependencies among their labels, which improves the quality of expert codification, as described in [4,38]. To provide further insights, we also generated visual explanations for a subset of the samples in the test dataset. To achieve this, Grad-CAM++ was applied to a given image and class K to determine the regions of the image that are most competitive for classification. As seen in the CAM, the important regions highlighted in red reveal that our model can effectively exploit the most relevant features to correctly identify the image’s class. The second row of the figure depicts some visual explanations on the 2D projections. Notably, class 4 can be confused by the model with a seven or a nine in a few samples.

In the last row of Figure 7, we can infer that our method effectively identifies the zones where the labelers have the best accuracy. This is not unexpected as the annotators’ accuracy (simulated) is compared with their reliability (estimated); therefore, the clusters where a specific labeler obtains the highest accuracy should align with the clusters where the estimated reliability is closest to 1 (yellow). To further support this statement, we depict the estimated probability function through a kernel density estimation (KDE) plot, to show the reliability estimated per annotator. For example, regarding annotator 1 (blue), as most of its estimates are reliable, the KDE increases when the reliability is 1. However, for annotator two (orange), its peak KDE value is slightly lower when the reliability is one. Similarly, annotators 3 (green) and 4 (red) exhibit an inverse behavior, as their performance is more doubtful.

In addition, it is important to note that our proposed GCECDL encodes the interdependence between annotators. By comparing the simulated annotators and the labelers’ performance in Figure 6 and Figure 7, it is clear that our proposed model closely follows the performance pattern of the labelers’ capabilities. Therefore, when a suitable labeler is presented, the model provides a high estimation for the same labeler in the zones where the labelers have the best accuracy. Conversely, when malicious labelers are present, the model reflects this in poor reliability estimation. This highlights how the ability to label and reliability estimation are closely related. Furthermore, it is worth noting that annotators with high uncertainty tend to have CAMs with more energy, which supports the aforementioned statement empirically.

### 5.2. Method Comparison Results for Multiple-Annotator Classification

Table 3 presents the results of our experiments on real-world datasets. The non-parametric Friedman test results for every quantitative assessment measure establish their statistical significance. The null hypothesis is rejected, indicating that all algorithms attain different performances as described in [58]. Additionally, the significance threshold is fixed at p<0.05. The DL-GOLD standard is not included in the test to compare state-of-the-art approaches exclusively. It is worth noting that most classification schemes exhibit a highly satisfactory performance for the datasets with simulated annotators, as reflected in the scores.

Our approach outperforms the selected state-of-the-art methods across most evaluation measures. For example, our proposal achieves the highest accuracy on 12 of 15 datasets, as shown in Table 3. Similarly, GCECDL outperforms tested strategies regarding BACC, NMI, and AUC for most datasets. This outcome is due to the fact that GCECDL properly codes annotators’ reliability by considering the correlations among their decision, even for noisy outputs. Remarkably, RCDNN reaches the second highest average performance. Indeed, it obtains the highest performance on CIFAR-10H, LabelMe, and Music. In general, RCDNN is similar to GCECDL, but it is significantly affected when the annotators’ performance is below 60%. Furthermore, it struggles with databases with a high class imbalance ratio. This behavior highlights that RCDNN is more susceptible to noisy or imbalanced scenarios.

Moreover, we observe that high NMI values for some datasets suggest that the labels provided by the annotators are consistent and reliable. Our GCECDL method effectively captured these dependencies. For example, in the MNIST dataset, a high NMI value of 97.39% is attained, indicating that the labels provided by the different annotators and the hidden ground truth prediction are highly consistent. In contrast, in the Cats vs. Dogs dataset, we observe a lower NMI value of 20.79%, pointing to more significant variability in the annotations and highlighting the challenges of dealing with noisy or inconsistent annotations.

Finally, the DL-CL(MW) approach outperforms the DL-MV scheme, which can be observed in the average ranking. Furthermore, it is worth noting that DL-CL(MW) includes the introduction of the CrowdLayer, which allows for training neural networks directly from multiple labels without encoding the annotators’ behavior. On the other hand, DL-MV presents the lowest performance among the studied methods. It can be explained by the fact that DL-MV is the most naive approach, and most annotators were simulated with a low level of expertise, negatively impacting the outcome of the majority voting strategy.

## 6. Conclusions

This article introduced a Generalized Cross-Entropy-based Chained Deep Learning model, termed GCECDL, to deal with multiple-annotator scenarios. Our method follows the ideas of [43,46], where each parameter is modeled in a multi-labeler likelihood by using the outputs of a deep neural network. Nonetheless, unlike [43]—where a CCE-based loss was used—we also introduced a noise-robust loss function based on GCE [42] as a tradeoff between MAE and CCE. Thus, GCECDL codes the non-stationary patterns of each annotator regarding the input space. We tested our approach for classification tasks using fully synthetic and real-world databases from well-known repositories, including structured data and images. According to the results, our GCECDL can achieve robust predictive properties for the used datasets defeating the selected state-of-the-art models. We attribute this behavior to the coupled MAE and CE within GCE, exploiting the symmetry property of MAE for softmax-based outputs and the L1-norm and cross-entropy tradeoff in weighting noisy annotations as a function of the input space. In addition, our chained architecture yields a self-regularization strategy within a DL framework that favors proper labeler’s reliability estimation and ground truth prediction. On the other hand, we created visual explanations using GradCam++ [30] to identify the most influential regions that demonstrate the model’s ability to correctly predict the hidden ground truth and assess the reliability of the annotators. Furthermore, using t-SNE [29] to project the extracted features onto a two-dimensional space allowed us to retain the spatial relationships from the original input space.

As future work, extending the GCECDL for regression tasks is a promising research area, as demonstrated by the model introduced in [35]. Our next step is to experiment with various activation functions and deeper convolutional and recurrent architectures to tackle complex tasks such as computer vision, natural language processing, and graph-based modeling [59]. Additionally, we plan to develop a model for identifying non-stationary patterns from multidomain input spaces holding noisy targets in an agricultural context, using multispectral imagery, climatic data, and infield data, instead of relying on annotators [60]. Furthermore, we plan to test more Explainable AI methods to provide deeper insight into our model performance [61,62,63], e.g., Layer-wise Relevance Propagation that captures both negative and positive relevance. Finally, actionable and explainable AI extensions based on our GCECDL would be an exciting research line [64].

## Figures and Tables

**Figure 1 sensors-23-03518-f001:**
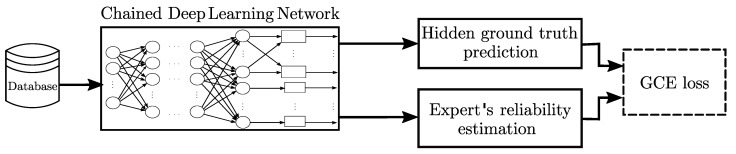
GCECDL sketch for multiple-annotator classification holding ground truth prediction and instance-based expert reliability estimation.

**Figure 2 sensors-23-03518-f002:**
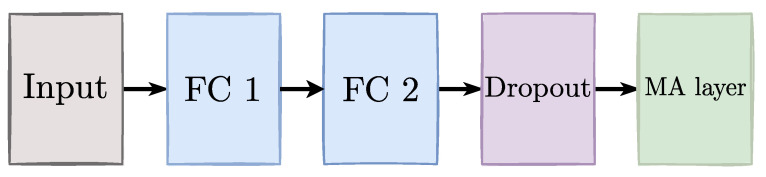
GCECDL-based fully connected architecture for tabular databases. FC stands for a fully connected (dense) layer. FC1 and FC2 use a selu activation. A dropout layer (Dropout) is included to avoid overfitting. The MA layer contains two fully connected layers that output the hidden ground truth label and the annotator’s reliability, fixing a softmax and a sigmoid activation, respectively.

**Figure 3 sensors-23-03518-f003:**
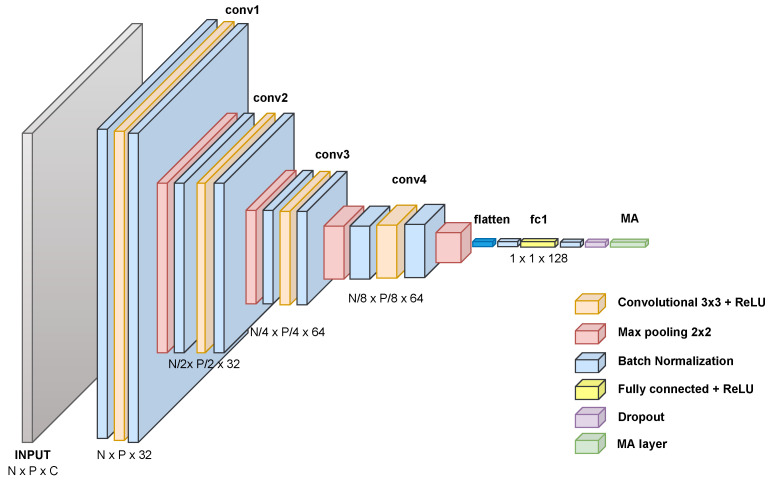
GCECDL-based convolutional architecture for image databases. Four convolutional layers with 3 × 3 patches, 2 × 2 max pooling, and ReLU activations are included. The MA layer outputs the hidden ground truth label and the annotator’s reliability fixing a softmax and a sigmoid activation, respectively.

**Figure 4 sensors-23-03518-f004:**
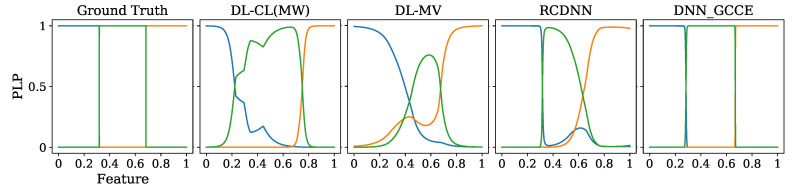
Synthetic results—1D dataset. The predictive label’s probability (PLP) is shown, comparing the prediction of our GCECDL (AUC = 0.99) against: DL-CL(MW) (AUC = 0.79), DL-MV (AUC = 0.9), and RCDNN (AUC = 0.99). Label classes are represented as blue, green, and orange.

**Figure 5 sensors-23-03518-f005:**
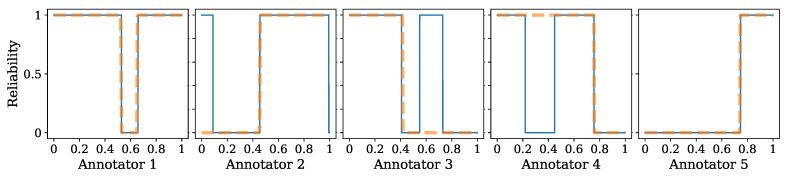
GCECDL-based annotators’ performance (reliability) estimation for the synthetic 1D experiment. Orange dashed line depicts (from left to right) the simulated accuracy for each annotator based on Equations (19)–(22). The blue line shows (from left to right) the estimated annotator’s reliability ($\lambda_{r}$).

**Figure 6 sensors-23-03518-f006:**
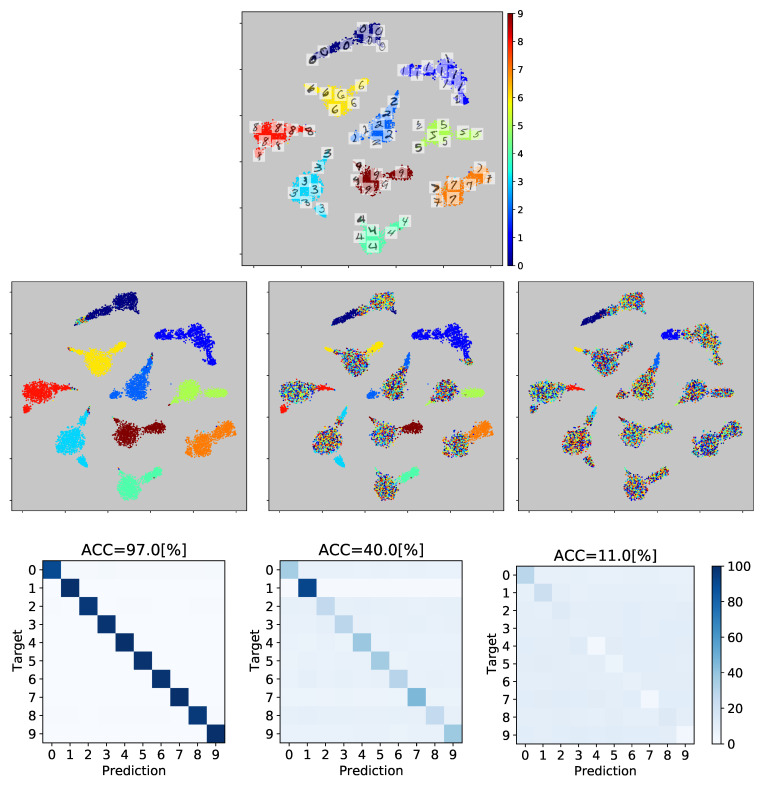
Simulated annotators’ for the MNIST database. A 2D t-SNE-based projection of the Mnist training set from the FC1 layer feature maps is shown (some exemplary images are depicted over their 2D projections, where numbers from zero to nine stand for each MNIST class). The ground truth labels are used as a color. The second row depicted the 2D projection of each annotator’s labels in color. The third row presents the confusion matrix of each expert and their achieved accuracy.

**Figure 7 sensors-23-03518-f007:**
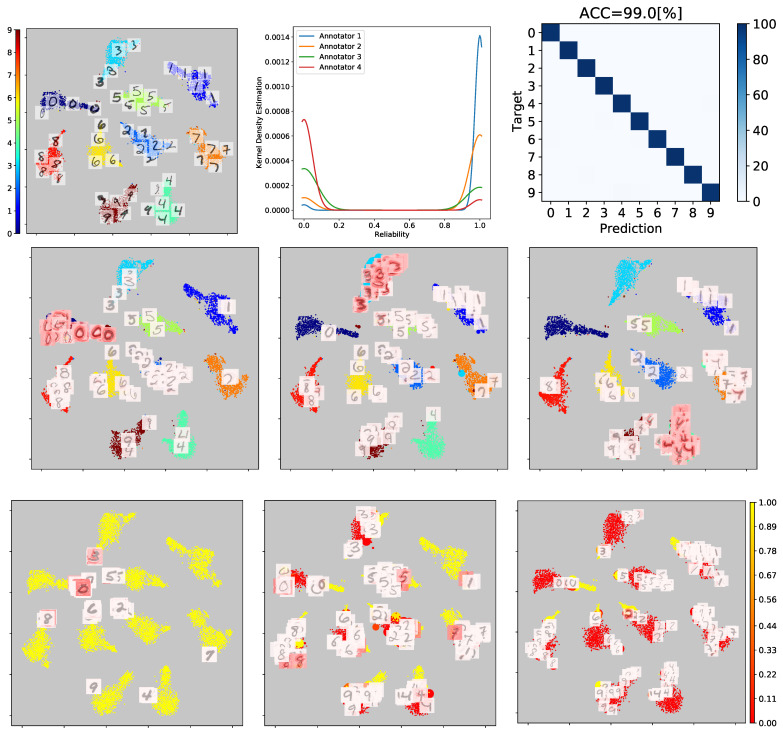
Visual inspection results for the MNIST database. A 2D t-SNE-based projection of the Mnist test set from the FC1 layer feature maps is shown (some exemplary images are depicted over their 2D projections, where numbers from zero to nine stand for each MNIST class). The ground truth labels are shown in color. The second row depicted the 2D projection of the ground truth prediction in color with some explanation maps for visualization purposes. The third row presents the 2D projection of each annotator’s reliability estimation in color.

**Table 1 sensors-23-03518-t001:** Description of the number of features (P), instances (N), and classes (K) of tested datasets for multiple-annotator classification.

Name	Input Shape (P)	Number of Instances (N)	Number of Classes (K)
1D Synthetic	1	500	3
Ocupancy	7	20,560	2
Skin	4	245,057	2
Tic-Tac-Toe	9	958	2
Balance	4	625	3
Iris	4	150	3
New Thyroid	5	215	3
Wine	13	178	3
Fashion-Mnist	784	70,000	10
Segmentation	18	2310	7
Western	7	3413	4
Cat vs. Dog	200 × 200 × 3	25,000	2
Mnist	28 × 28 × 1	70,000	10
CIFAR-10H	32 × 32 × 3	19,233	10
LabelMe	256 × 256 × 3	2688	8
Music	124	1000	10

**Table 2 sensors-23-03518-t002:** A brief overview of the state-of-the-art methods tested.

Algorithm	Description
DL-GOLD	A DL classification model using the real labels (upper bound).
DL-MV	A DL classification model using the MV of the labels as the ground truth.
RCDNN [43]	A regularized chained deep neural network which predicts the ground truth and annotators’ performance from input space samples.
DL-CL(MW) [28]	A crowd Layer for DL, where annotators’ parameters are constant across the input space.

**Table 3 sensors-23-03518-t003:** Method comparison results—multiple-annotator classification. Bold: the highest score for each measure excluding the upper bound (target) classifier DL-GOLD. For our GCECDL, the fixed q-value is presented in parentheses. Friedman test: we obtain a Chi-square of 27.08, 23.48, 28.03, and 7.92 (*p*-value = $5.6\times 10^{-6}$, $3.56\times 10^{-6}$, $3.20\times 10^{-5}$, and 0.047) for ACC, BACC, NMI, and AUC measures, respectively.

Database	Measure	DL-GOLD	DL-CL(MW)	DL-MV	RCDNN	GCECDL
Ocupancy	ACC [%]	97.59 ± 0.11	90.89 ± 0.92	90.31 ± 0.80	96.61 ± 0.42	**97.50** ± **0.06 (0.1)**
	BACC [%]	95.92 ± 0.09	74.30 ± 3.04	75.50 ± 0.99	93.22 ± 1.27	**95.86** ± **0.10 (0.1)**
	NMI [%]	83.91 ± 0.35	57.98 ± 2.16	52.32 ± 3.56	77.93 ± 2.51	**83.64** ± **0.31 (0.1)**
	AUC [%]	97.96 ± 0.05	87.17 ± 1.52	87.75 ± 0.49	99.03 ± 0.14	**97.92** ± **0.05 (0.1)**
Skin	ACC [%]	99.74 ± 0.13	56.48 ± 0.36	80.55 ± 3.23	89.45 ± 2.06	**96.06** ± **0.59 (0.1)**
	BACC [%]	99.62 ± 0.15	45.07 ± 0.46	79.80 ± 1.79	64.27 ± 10.66	**90.60** ± **4.31 (0.1)**
	NMI [%]	96.86 ± 1.15	18.31 ± 0.21	13.56 ± 1.06	38.99 ± 8.25	**71.14** ± **5.22 (0.1)**
	AUC [%]	99.81 ± 0.08	91.80 ± 1.03	80.45 ± 6.22	82.13 ± 5.33	**99.30** ± **0.10 (0.1)**
Tic-Tac-Toe	ACC [%]	98.47 ± 0.17	80.00 ± 3.73	78.26 ± 1.85	90.49 ± 17.16	**97.28** ± **1.17 (0.2)**
	BACC [%]	96.00 ± 0.27	66.24 ± 5.98	56.09 ± 2.14	78.74 ± 38.93	**93.10** ± **2.78 (0.2)**
	NMI [%]	88.57 ± 1.52	36.88 ± 6.07	23.90 ± 2.05	71.42 ± 24.79	**83.52** ± **6.13 (0.2)**
	AUC [%]	98.05 ± 0.14	96.48 ± 1.00	85.37 ± 1.21	91.71 ± 21.93	**96.55** ± **1.39 (0.2)**
Balance	ACC [%]	91.91 ± 2.50	85.27 ± 1.93	73.72 ± 1.51	**91.33** ± **1.14**	90.21 ± 0.83 (0.1)
	BACC [%]	60.00 ± 5.68	74.9 ± 4.09	43.39 ± 0.8	46.93 ± 1.00	**75.27** ± **7.33 (0.1)**
	NMI [%]	73.34 ± 3.32	61.98 ± 3.58	59.43 ± 2.04	66.52 ± 3.30	**68.00** ± **2.19 (0.1)**
	AUC [%]	83.66 ± 4.91	**84.97** ± **4.08**	70.66 ± 3.35	64.51 ± 17.36	81.11 ± 8.25 (0.1)
Iris	ACC [%]	97.34 ± 0.94	95.11 ± 2.59	95.33 ± 1.59	96.89 ± 1.15	**97.34** ± **0.94 (0.1)**
	BACC [%]	96.47 ± 1.24	91.67 ± 4.54	91.56 ± 1.84	94.24 ± 2.03	**96.06** ± **2.11 (0.1)**
	NMI [%]	91.69 ± 1.91	86.82 ± 5.59	87.61 ± 2.86	89.68 ± 3.74	**91.15** ± **3.07 (0.1)**
	AUC [%]	100.00 ± 0.00	100.00 ± 0.00	100.00 ± 0.00	99.83 ± 0.00	**100.00** ± **0.00 (0.1)**
New Thyroid	ACC [%]	96.15 ± 0.77	95.69 ± 1.15	**97.69** ±**1.03**	95.08 ± 0.62	96.00 ± 1.84 (0.01)
	BACC [%]	92.5 ± 1.94	92.61 ± 2.8	**93.88** ±**2.65**	88.41 ± 2.22	90.52 ± 4.36 (0.01)
	NMI [%]	82.98 ± 3.44	82.38 ± 3.86	**90.44** ±**3.61**	80.40 ± 2.42	83.61 ± 6.85 (0.01)
	AUC [%]	96.19 ± 0.96	95.46 ± 1.78	**97.80** ± **0.6**	94.72 ± 1.67	95.19 ± 2.23 (0.01)
Wine	ACC [%]	97.59 ± 0.85	95.74 ± 1.67	73.15 ± 3.73	94.26 ± 2.10	**96.48** ± **2.38 (0.01)**
	BACC [%]	96.59 ± 1.08	94.17 ± 2.52	64.44 ± 5.42	91.99 ± 2.59	**96.68** ± **2.92 (0.01)**
	NMI [%]	91.50 ± 3.07	85.54 ± 4.03	44.18 ± 4.39	82.91 ± 5.97	**89.87** ± **5.37 (0.01)**
	AUC [%]	98.88 ± 0.37	99.88 ± 0.37	87.76 ± 2.88	97.13 ± 1.26	**100.0** ± **0.00 (0.01)**
F-Mnist	Acc [%]	86.45 ± 0.25	79.54 ± 9.16	85.83 ± 0.35	86.98 ± 0.71	**88.26** ± **0.26 (0.01)**
	BACC [%]	84.92 ± 0.27	77.28 ± 10.11	84.23 ± 0.38	85.50 ± 0.77	**86.94** ± **0.28 (0.01)**
	NMI [%]	77.16 ± 0.21	72.92 ± 3.84	76.46 ± 0.38	78.75 ± 0.48	**79.89** ± **0.21 (0.01)**
	AUC [%]	88.75 ± 1.61	85.61 ± 13.95	90.42 ± 1.46	89.33 ± 2.93	**89.11** ± **2.27 (0.01)**
Segmentation	ACC [%]	95.17 ± 0.25	94.65 ± 0.95	91.11 ± 0.94	91.85 ± 0.67	**94.88** ± **0.8 (0.01)**
	BACC [%]	94.49 ± 0.59	93.81 ± 1.09	89.79 ± 1.09	90.69 ± 0.77	**94.15** ± **0.95 (0.01)**
	NMI [%]	91.07 ± 0.77	**90.38** ±**1.18**	84.46 ± 1.65	86.27 ± 0.70	89.92 ± 1.25 (0.01)
	AUC [%]	99.95 ± 0.13	99.76 ± 0.20	98.71 ± 0.44	98.94 ± 0.33	**99.79** ± **0.22 (0.01)**
Western	ACC [%]	99.25 ± 0.38	95.94 ± 4.39	95.11 ± 1.15	97.89 ± 0.40	**98.31** ± **0.61 (0.01)**
	BACC [%]	99.22 ± 0.42	94.37 ± 5.54	93.47 ± 1.45	97.25 ± 0.57	**98.00** ± **0.97 (0.01)**
	NMI [%]	98.61 ± 0.70	96.80 ± 1.70	94.33 ± 0.97	97.16 ± 0.30	**97.35** ± **0.60 (0.01)**
	AUC [%]	100.00 ± 0.00	100.00 ± 0.00	98.97 ± 0.85	99.99 ± 0.01	**100.00** ± **0.00 (0.01)**
Cats vs. Dogs	ACC [%]	85.00 ± 0.44	68.97 ± 0.84	58.69 ± 1.02	70.84 ± 4.71	**74.15** ± **0.58 (0.1)**
	BACC [%]	70.19 ± 0.22	37.44 ± 0.35	16.50 ± 0.51	43.51 ± 2.08	**48.24** ± **0.54 (0.1)**
	NMI [%]	40.66 ± 0.41	11.03 ± 0.25	2.50 ± 0.12	16.02 ± 1.04	**20.79** ± **0.33 (0.1)**
	AUC [%]	93.17 ± 0.22	68.94 ± 0.80	58.63 ± 0.98	70.98 ± 4.57	**82.99** ± **0.30 (0.1)**
Mnist	ACC [%]	99.32 ± 0.06	87.99 ± 2.73	92.88 ± 0.54	99.09 ± 0.05	**99.11** ± **0.08 (0.01)**
	BACC [%]	99.22 ± 0.06	86.49 ± 3.11	91.97 ± 0.59	98.98 ± 0.06	**99.02** ± **0.09 (0.01)**
	NMI [%]	97.95 ± 0.15	82.82 ± 2.29	83.43 ± 1.02	97.28 ± 0.11	**97.39** ± **0.21 (0.01)**
	AUC [%]	99.81 ± 0.08	99.82 ± 0.02	97.88 ± 0.33	99.71 ± 0.06	**99.68** ± **0.08 (0.01)**
CIFAR-10H	ACC [%]	71.72 ± 1.12	60.80 ± 1.59	68.24 ± 1.05	**69.53** ± **0.63**	69.24 ± 0.67 (0.01)
	BACC [%]	68.46 ± 0.28	56.56 ± 1.72	64.69 ± 1.18	**66.18** ± **0.72**	65.88 ± 0.76 (0.01)
	NMI [%]	64.2 ± 0.23	43.09 ± 1.21	49.55 ± 1.4	**51.44** ± **0.81**	50.95 ± 0.78 (0.01)
	AUC [%]	96.08 ± 0.11	89.42 ± 0.7	94.81 ± 0.35	**95.17** ± **0.16**	95.01 ± 0.19 (0.01)
LabelMe	ACC [%]	90.91 ± 0.44	83.11 ± 0.96	76.94 ± 1.15	**89.09** ± **0.41**	88.97 ± 0.55 (0.01)
	BACC [%]	90.03 ± 0.42	81.85 ± 0.86	75.08 ± 1.25	**88.01** ± **0.42**	87.97 ± 0.57 (0.01)
	NMI [%]	81.36 ± 0.79	73.92 ± 0.80	68.36 ± 0.54	77.82 ± 0.55	**78.00** ± **0.70 (0.01)**
	AUC [%]	99.37 ± 0.06	96.34 ± 0.88	97.65 ± 0.13	**99.14** ± **0.06**	99.11 ± 0.08 (0.01)
Music	ACC [%]	76.4 ± 0.88	57.13 ± 2.80	62.8 ± 1.06	**65.80** ± **2.83**	65.63 ± 2.41 (0.01)
	BACC [%]	74.32 ± 0.91	51.47 ± 2.91	57.67 ± 1.27	**61.96** ± **3.32**	61.83 ± 2.84 (0.01)
	NMI [%]	65.86 ± 1.16	57.25 ± 1.55	53.87 ± 1.49	57.02 ± 2.63	**57.36** ± **1.40 (0.01)**
	AUC [%]	96.34 ± 0.13	88.62 ± 3.08	93.58 ± 0.21	94.07 ± 0.94	**94.13** ± **0.50 (0.01)**
Average Ranking	ACC [%]	−−±−−	3.26 ± −−	3.40 ± −−	2.00 ± −−	**1.33** ± −−
	BACC [%]	−−±−−	3.00 ± −−	3.40 ± −−	2.33 ± −−	**1.26** ± −−
	NMI [%]	−−±−−	3.14 ± −−	3.46 ± −−	2.21 ± −−	**1.20** ± −−
	AUC [%]	−−±−−	2.67 ± −−	3.13 ± −−	2.47 ± −−	**1.73** ± −−

## Data Availability

Publicly available datasets were analyzed in this study. These data can be found at the UCI repository (http://archive.ics.uci.edu/ml, (accessed on 19 August 2022)), Keel-dataset repository (https://sci2s.ugr.es/keel/category.php?cat=clas, (accessed on 3 October 2022)), and Rodrigues repository (http://www.fprodrigues.com/, (accessed on 20 Sepetember 2022)). Fashion-Mnist is freely available at dataset repository (https://github.com/zalandoresearch/fashion-mnist, (accessed on 27 August 2022)).

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
