# Peer review of "Chained Deep Learning Using Generalized Cross-Entropy for Multiple Annotators Classification"

_sensors, 2023, doi:10.3390/s23073518_

Round 1
Reviewer 1 Report
Generally speaking, the problem proposed in this paper is interesting, and the obtained results are promising and correct. The references in this paper are appropriate and most of the references cited by the author are recent ones.
It is enough to support the research in this paper. Nevertheless, there are some problems that should be addressed.
The abstract size is very large. I think the maximum number of words in the abstract must be less than 300, please check it.
The article has no literature Review Section in the paper. Please add a Literature review section in the paper to prove that before and during the writing of the paper, the authors have studied the concerned and the latest research.
I would suggest if it could be possible, some simulation comparison with one of the other techniques proposed in the technical literature and disclose the superiority of your work concerning the literature.
The paper has many typo mistakes. Please revisit all the paper and add/remove spaces.
The results are required to be placed under the heading of the Results and Discussion Section. Please discuss each result properly.
Reviewer 2 Report
This paper describes a new method that is invented to tackle the problems arising from unreliable or different human annotations on data. When datasets are used for training Machine Learning (ML) models, human-labelled data are expected to be perfect. In reality, humans do not have the same level of expertise or background knowledge and can disagree when labelling a data sample. The goal of this work is to take into account the variability of labels and dependencies between humans (as far as labelling is concerned) and at the same time the model architecture and its loss function. The resulting model is trained on several datasets of different types with a simulated labelling procedure and the method has shown its adequacy and better performance in comparison to other methods.
The paper contains good content to justify publication. Nevertheless, the reviewers have concerns about the fact that the labelling is simulated and it is questionable if this approximation is valid. If the authors provided experiments with human data that were near enough to the simulated ones, then the reviewers could conclude that this work is of great value. To validate its importance and its execution, the GitHub repository is also necessary.
The related work and theory have a good enough range. Nevertheless, some important references are missing that deal mainly with Explainable AI (xAI) methods. The use of only one xAI method – namely Grad-CAM++ - is not enough to provide insight for all those cases. In particular, it has been shown that Layer-wise Relevance Propagation (LRP) has substantial benefits since it captures both negative and positive relevance. On page 12 the high energy region with the CAM explanation is not necessarily because of high uncertainty, therefore other methods are needed as well. For more explanations about how xAI methods work, the following references are helpful:
- Bennetot, A., Donadello, I., Qadi, A. E., Dragoni, M., Frossard, T., Wagner, B., ... & Díaz-Rodríguez, N. (2021). A Practical Tutorial on Explainable AI Techniques. arXiv preprint arXiv:2111.14260.
https://doi.org/10.2139/ssrn.4229624
- Holzinger, A., Saranti, A., Molnar, C., Biecek, P., & Samek, W. (2022). Explainable AI methods-a brief overview. In International Workshop on Extending Explainable AI Beyond Deep Models and Classifiers (pp. 13-38). Springer, Cham.
https://doi.org/10.1007/978-3-031-04083-2_2
It is important to note that nowadays it is not advisable to just present large tables of metric results to show the adequacy of a new method. To show how one could improve the labelling procedure from the insights gained, Actionable xAI (AxAI) was invented as a practice:
- Saranti, A., Hudec, M., Mináriková, E., Takáč, Z., Großschedl, U., Koch, C., ... & Holzinger, A. (2022). Actionable Explainable AI (AxAI): A Practical Example with Aggregation Functions for Adaptive Classification and Textual Explanations for Interpretable Machine Learning. Machine Learning and Knowledge Extraction, 4(4), 924-953.
https://doi.org/10.3390/make4040047
It would be very good as a future work direction to use the results of this paper for targeted suggestions to human annotators.
This research work is based on a solid methodology in general. Issues that could be improved include first the non-depiction of non-stationarity patterns of each annotator on page 3 and page 6 as well as the nature of the assumed interdependence among labellers/humans. This would help enormously for argumentation for this method. The reviewers advise on page 4 a little more explanation of the noise robustness and symmetry of the MAE metric. Furthermore, it is not clear what is being fixed in section 2.2. and how the function composition presented on page 11 can be implemented for Fully-Connected Neural Networks (FCNN) and Convolutional Neural Networks (CNN) that are being used in this paper. The lack of a concrete example shows bad scientific practice; the reviewers require it for the revised version.
It is highly appreciated that the researchers used many different datasets of different types to make many experiments. Training with an imbalanced dataset should not be measured by the accuracy metric at all (section 3.3. on page 7) – please check section 44.5. of the following book and provide the results of Mutual Information (MI) instead of accuracy:
- MacKay, D. J., & Mac Kay, D. J. (2003). Information theory, inference and learning algorithms. Cambridge university press.
https://doi.org/10.1017/s026357470426043x
It is good that the ROC metric is provided. On page 9 the results of the combination matrix and equation (21) are not analyzed. What does this combination matrix tells/shows the reader? For Figure 4 MI should also be used, or at least the ROC metric. It is not clear if in Figure 3 a well-known architecture is depicted and since the researchers refer that they might need a better one on page 12 section 4.2. it is unclear why they didn’t use one – since Tensorflow provides a plethora of them as well as pre-training and augmentation methods. The researchers must present the “difficult” cases on page 10, but it is not clear why this happens – xAI methods, as mentioned above, can help uncover these issues. The reviewers suggest also that the results of Figures 6 and 7 are compared with the results of corresponding neural networks that are trained with agreeable human labels to show the fundamental differences. Is there any possibility that this research work is used for the creation of guidelines for conducting experiments for human annotations? It is highly advised that the users use it in their future work graph data and Graph Neural Networks for analysis purposes.
The paper is well-organized and written with only minor typos. In figure 1 it is advised to provide an explanation and name for the rectangular elements in the last layer of the network, that even have different sizes. The losses could also be depicted in this figure.
Reviewer 3 Report
The authors propose a Generalized Cross-Entropy-based framework using Chained Deep Learning (GCECDL) that uses a deep learning-based
approach to building a supervised learning model in the context of multiple annotator classification. The proposed approach has been supported by extensive experiments comparing its performance and validity to that of state-of-the-art methods.
Reviewer 4 Report
Abstract
Add a brief conclusion
Introduction
L67: Please extend cites [17 - 19] to [17, 18, 19]
L97: DL means deep learning, and if so, extend since the meaning of said acronyms is not mentioned before.
Explain briefly which problem solves the proposed method.
Methods
How about split ratio of training/testing/validation?
It´s possible to mention the time it takes your GCECDL in the classification.
Results
Add a table comparing you GCECDL accurry vs othe similar classification methods.
You can supplement according to table 1 which database your method has a better improvement.
Conclusions
L395: Can you explain the attributtion of robustness if necessary substantiate.
Round 2
Reviewer 2 Report
The revised version shows improvements. The reviewers approve the use of mutual information and the future work ideas the most.